# A Vaccine for Canine Rocky Mountain Spotted Fever: An Unmet One Health Need

**DOI:** 10.3390/vaccines10101626

**Published:** 2022-09-28

**Authors:** David H. Walker, Lucas S. Blanton, Maureen Laroche, Rong Fang, Hema P. Narra

**Affiliations:** 1Department of Pathology, The University of Texas Medical Branch at Galveston, 301 University Blvd., Galveston, TX 77555-0609, USA; 2Department of Internal Medicine, Division of Infectious Diseases, The University of Texas Medical Branch at Galveston, 301 University Blvd., Galveston, TX 77555-0435, USA; 3Department of Microbiology and Immunology, The University of Texas Medical Branch at Galveston, 301 University Blvd., Galveston, TX 77555-1019, USA

**Keywords:** Rocky Mountain spotted fever, One Health, *Rickettsia rickettsii*, *Rhipicephalus sanguineus*, canine vector-borne disease, subunit vaccine, live attenuated vaccine

## Abstract

Outbreaks of life-threatening Rocky Mountain spotted fever in humans and dogs associated with a canine-tick maintenance cycle constitute an important One Health opportunity. The reality of the problem has been observed strikingly in Mexico, Brazil, Colombia, and Native American tribal lands in Arizona. The brown dog tick, *Rhipicephalus sanguineus sensu lato*, acquires the rickettsia from bacteremic dogs and can maintain the bacterium transtadially to the next tick stage. The subsequent adult tick can then transmit infection to a new host, as shown by guinea pig models. These brown dog ticks maintain spotted fever group rickettsiae transovarially through many generations, thus serving as both vector and reservoir. Vaccine containing whole-killed *R. rickettsii* does not stimulate sufficient immunity. Studies of *Rickettsia* subunit antigens have demonstrated that conformationally preserved outer-membrane autotransporter proteins A and B are the leading vaccine candidates. The possibility of a potentially safe and effective live attenuated vaccine has only begun to be explored as gene knockout methods are applied to these obligately intracellular pathogens.

## 1. Introduction

One Health encompasses an integrated unifying approach to balance and optimize the health of three components: people, animals, and the environment. As cited in the Centers for Disease Control and Prevention website, “The health of people is closely connected to the health of animals and the environment”. In the setting of outbreaks of severe Rocky Mountain spotted fever involving many persons in a particular environment, dogs and *Rhipicephalus sanguineus* ticks in homes play critical roles, and control of the zoonotic cycle involves strategies to interrupt the maintenance and transmission of *Rickettsia rickettsii*.

## 2. The Current Status of the One Health Canine-Brown Dog Tick-*Rickettsia rickettsii* Threat

Dogs are susceptible to Rocky Mountain spotted fever. The disease can be fatal in some breeds of dogs, as documented in natural outbreaks [1] and clinical and experimental studies [2,3,4,5,6]. Epidemiologic studies point to brown dog ticks, *Rhipicephalus sanguineus*, as the likely vectors of outbreaks of Rocky Mountain spotted fever in Mexico [7,8,9,10,11,12,13,14,15,16,17,18] and in Native American tribal lands in Arizona [19,20,21].

Vaccination of dogs to prevent their infection with *Rickettsia rickettsii* has received little attention. A recent study determined that, although a heat-killed whole *R. rickettsii* vaccine prepared from organisms propagated in embryonated chicken eggs and purified by differential centrifugation given with Montanide adjuvant and boosted once protected dogs from death, four of six animals developed a brief fever, one developed a mild rash, and three had documented rickettsemia [22]. A recombinant peptide vaccine of OmpB encoded by nucleotides 2638–3822 did not protect dogs [22].

*Rickettsia rickettsii* vaccines were developed and used in humans. The original vaccine developed at the Rocky Mountain Laboratory in 1924 from the tissues of infected *Dermacentor andersoni* ticks reduced the fatality rate of vaccinees but did not prevent illness [23,24]. The subsequent breakthrough discovery that *R. rickettsii* could be propagated in yolk sac of embryonated chicken eggs altered the dangerous method of propagating in infected ticks to chicken egg cultivation of *R. rickettsii* for vaccine production in 1938 [25]. Subsequent human infectious *R. rickettsii* challenge studies demonstrated that neither vaccine prevented infection [26]. The US Army developed a cell-culture-propagated whole-killed *R. rickettsii* vaccine that when tested in human volunteers prevented only 25% of vaccinees from developing Rocky Mountain spotted fever [27,28]. The use of a whole-killed *R. rickettsii* vaccine would not seem to be effective in preventing the transmission of rickettsiae to dogs or achieve the goal of interrupting the *Rh. sanguineus*-dog cycle of maintenance of *R. rickettsii* in the environment.

## 3. Clinical and Epidemiological Significance of Rocky Mountain Spotted Fever

Rocky Mountain spotted fever (RMSF), caused by *Rickettsia rickettsii*, is a severe, life-threatening tick-borne infection. Disease is characterized by fever, headache, myalgia, and rash [29]. Although prompt administration of tetracycline antibiotics, such as doxycycline, is quite effective when given during the first 5 days of illness [30], the disease is often difficult for clinicians to recognize. There is no point-of-care diagnostic test to accurately diagnose RMSF. Furthermore, signs and symptoms are largely undifferentiated and may be erroneously attributed to a wide variety of bacterial (e.g., meningococcus, syphilis, endocarditis, leptospirosis, typhoid fever), viral (e.g., roseola, measles, rubella, mononucleosis, acute HIV, dengue fever) and non-infectious syndromes (e.g., drug eruptions, Kawasaki disease, vasculitides) [31]. When effective empiric therapy is not initiated early in illness, there is greater likelihood of unfavorable outcomes [32,33]. Examples of severe manifestations include respiratory failure requiring intubation and ventilatory support [34], acute kidney injury requiring hemodialysis, gangrenous digits that may require amputation [35,36], and meningoencephalitis, which can lead to longstanding or permanent neurologic sequelae (seizures, cognitive dysfunction, behavioral abnormalities, stroke, and coma) [36,37]. In the preantibiotic era, the case fatality rate in the United States approached 23% [38]. Despite declines in the fatalities after the availability of chloramphenicol and tetracyclines, the case fatality rate has been reported to be up to 5% in the postantibiotic era [30], and in tribal lands of Arizona, the case fatality rate has been reported to be up to 10% [39]. The potential severity of RMSF highlights its clinical importance.

In the United States, RMSF is classically transmitted by the bite of *Dermacentor andersoni* ticks in the western mountainous states and *D. variabilis* in the eastern half of the country [40]. Sixteen cases of RMSF were reported from rural tribal lands in Arizona in 2005 [19]. Two of these patients, both young children, died [19,41]. Prior to these 16 cases (diagnosed between 2003 and 2004) there were only 3 cases of RMSF reported in the entire state of Arizona in the preceding two decades [42,43]. This emergence of RMSF in Arizona was attributed to *R. rickettsii*-infected *Rhipicephalus sanguineus*—the ubiquitous brown dog tick [19]. Tribal lands of Arizona are now highly endemic [32], and in a study of RMSF in Native Americans, the Arizona outbreak was shown to affect children at higher rates than adults and has been associated with greater frequency of hospitalizations compared to other areas of the U.S. [44]. The outbreak in Arizona highlights the importance of the brown dog tick to the epidemiology of RMSF and resurfaces aspects of the disease that have been described in Latin America. 

The history of RMSF outbreaks in Mexico is tied closely to dogs and brown dog ticks, as detailed in a review by Álvarez-Hernández and colleagues [7]. In the 1920s, a fatal disease that resembled RMSF was described in the state of Sinaloa and was associated with *Rh. sanguineus*-infested dogs. In 1943, Mexican epidemiologists detailed a lethal disease (fatality rate of 80%) occurring in rural areas of the states of Sinaloa and Sonora, which would be recognized as RMSF. Again, the association with *Rh. sanguineus* and dogs was evident. In addition to infested dogs, the peridomiciliary behavior of these ticks was noted. Carried into homes by dogs, these ticks were frequently found in earthen walls of homes, on flooring, and in the bedding of those afflicted with RMSF. Later, in the early 1950s, an outbreak of RMSF in the La Laguna region of Mexico was attributed to *Rh. sanguineus* as the culprit vector [7]. After decades with few cases, a phenomenon hypothesized to be caused by DDT (dichlorodiphenyltrichoroethane) use [45], RMSF has reemerged in Mexico during the last two decades [7]. Contemporary outbreaks have again been associated with tick-infested free roaming dogs and have occurred in La Laguna, Sonora, and Baja California, where the reported case fatality rate in the latter two regions is 44% and 18%, respectively [16,46].

RMSF outbreaks have occurred in Brazil, where it is called Brazilian spotted fever, since the 1920s. Cases decreased in the 1950s and remained low until the 1980s, when the disease alarmingly reemerged [47]. Most cases are reported in southeastern Brazil (the highest incidence is in São Paulo state) [48], and the case fatality rate is as high as 40% [49]. The ecology of *R. rickettsii* in Brazil is complex. *Amblyomma* species are the main vectors, and capybaras (a large rodent) serve as an amplifying mammalian reservoir. Capybaras have encroached on suburban habitats, putting them in close contact with human populations [48]. When dogs cross the environment in which capybaras live, they can serve as bridge hosts to facilitate human contact with infected ticks [50]. Indeed, dogs have been documented to have a relatively high seroprevalence to spotted fever group rickettsiae in Brazil (up to 70%) [51,52,53], and as many as 13% of *Rh. sanguineus* have been found to be infected with *R. rickettsii* [51]. Additionally, three dogs belonging to a family of a child who died of RMSF in São Paulo demonstrated seroreactivity to *R. rickettsii*. In addition, *R. rickettsii*-infected *A. aureolatum* ticks were detected from one of these dogs [54]. Thus, dogs may play a role in the ecology of *R. rickettsii* in Brazil.

An outbreak of RMSF occuring in Tobia, Colombia was reported in 1937 and was remarkable for a 95% case fatality rate [55]. The disease was forgotten until two cases were identified in 2003 in Villeta, Colombia—a community near Tobia [56]. Although the tick species associated with these cases is unknown, one of the deaths in Villeta was a pregnant woman whose family had three dogs with an illness believed to resemble canine RMSF (two of these dogs died and one recovered with doxycycline therapy) [56]. Outbreaks in Colombia have been subsequently described in Cordoba and Antioquia [57]. The latter description was of a patient cluster, which was linked to the presence of *Rh. sanguineus* and *R. rickettsii* seropositive dogs [58].

## 4. Brown Dog Ticks as Vectors of *Rickettsia rickettsii*

The brown dog tick, *Rhipicephalus sanguineus sensu lato* (Acari: Ixodidae), is a cosmopolitan tick commonly found on dogs. The taxonomy of this group is still unresolved, but it is considered to be a species group that includes two described lineages—tropical and temperate—associated with distinct vectorial capacities [59]. *Rhipicephalus sanguineus s.l.* ticks are involved in the maintenance and transmission of pathogens of high medical and veterinary importance such as *Rickettsia* spp. [60,61,62,63].

The association between *Rh. sanguineus s.l.* and *R. rickettsii* has been studied mostly in Latin America, using molecular methods. In Mexico, two studies reported 31% [15] and 26% [64] prevalence rates of *R. rickettsii* in *Rh. sanguineus s.l.* ticks collected in Mexicali and Yucatan state, respectively. In Coahuila state in northern Mexico, pools of ticks were tested by molecular methods, revealing a minimum infection rate of 3.3%. In Brazil, reported prevalence rates range from 1.27% in São Paulo [53] to 13.1% in Juiz de Fora [51]. Finally, a study from Panama City, Panama, reported that 8.7% of *Rh.*
*sanguineus s.l.* ticks collected in the vicinity of RMSF cases were positive for *R. rickettsii*.

A few studies have been conducted in the United States. In the early 2000s, ticks were collected in Arizona from the homes of RMSF patients with history of contact with tick-infested dogs or/and tick bites. *Rickettsia rickettsii* was detected and isolated from 2.2% of collected ticks [19]. Finally, during a surveillance study in Riverside County, California, one *Rh. sanguineus s.l.* tick collected from the vegetation contained *R. rickettsii* [65].

Dogs are the preferential host of *Rhipicephalus sanguineus* ticks [66]. Increased temperatures make these ticks more opportunistic. They will feed on available hosts such as other domestic animals, small wild mammals, including cats, hedgehogs, and rodents, or humans. Ticks exposed to 40 °C attached much more avidly to human skin than those maintained at room temperature, and ticks showed a greater preference for humans at higher temperatures [67,68]. The low host-specificity of this tick increases the risk of transmission of vector-borne pathogens.

Very few studies have investigated the role of *Rh. sanguineus s.l.* in the transmission of *R. rickettsii*. Piranda et al. showed that *Rh. sanguineus s.l.* nymphs and larvae acquire *R. rickettsii* from bacteremic dogs and that a third of the ticks maintained the bacterium to the next stage. Moreover, the subsequent adult ticks are able to infect guinea pigs, which develop high fever, rickettsiae in splenic samples, and rickettsiae in blood [60]. The role of *Rh. sanguineus s.l.* in the transmission of *R. rickettsii* is further supported by the findings of Silva Costa et al., who demonstrated that the bacterium invades the tick ovaries during infection. In their study, *R. rickettsii* successfully invaded the oocytes of both semi-engorged and fed-to-repletion female *Rh. sanguineus* ticks [69]. Migration of pathogens to the salivary glands and ovaries of arthropod vectors is a crucial step for horizontal and transovarian transmission, respectively. It is therefore significant that the demonstration of the presence of live pathogens in these anatomic locations is a strong suggestion that the arthropod might be a vector, reservoir, and an element in an important zoonotic cycle.

*Rhipicephalus* ticks are associated with several rickettsiae other than *R. rickettsii*, and their transmission has been more studied. *Rickettsia conorii*, for example, is the etiologic agent of Mediterranean spotted fever (MSF), a rickettsial disease endemic in northern Africa and Southern Europe [70]. *Rhipicephalus sanguineus* s.l. ticks were implicated in the transmission of this *Rickettsia* spp. a century ago when Blanc and Campinopetros demonstrated in 1932 that patients inoculated with crushed infected ticks contracted MSF [61]. Transovarial transmission *of R. conorii* was confirmed using naturally infected ticks collected from dogs in the vicinity of MSF cases. The ticks were maintained under laboratory conditions, and eggs and larvae were tested after each generation, up to 11 generations. At the eleventh generation, the bacterium was still detected in all ticks tested and, in this study, *R. conorii* was also shown to invade the salivary glands and ovaries of *Rh. sanguineus s.l.* [71].

*Rhipicephalus turanicus* is a tick species closely related to *Rh. sanguineus s.l*, which is highly similar in terms of morphology and genotypic features. Both species commonly infest dogs [72,73]. *Rickettsia massiliae* is a spotted fever group *Rickettsia* spp. first isolated in 1990 from *Rh. turanicus*, near Marseille, France [74]. Transovarial transmission of the bacterium was further demonstrated by PCR testing and Gimenez staining of rickettsiae in hemolymph over several generations. High-fidelity transovarial transmission (100%) was demonstrated as well as the presence of the bacterium in the tick saliva [75].

Although the literature documenting the role of *Rh. sanguineus s.l.* in the transmission of *R. rickettsii* is incomplete, there is enough evidence to support the conclusion that *Rh. sanguineus* ticks are efficient vectors and reservoirs of rickettsiae. Targeting their main host, dogs, would be a highly relevant approach to control vector-borne pathogens transmitted by these ticks.

Considering the role of dogs in transmitting RMSF to humans, an ideal vaccine against *R. rickettsii* in canines would not only be safe but also, probably more importantly, confer sterile immunity without active bacterial infection in blood.

## 5. Potential Vaccine Strategies

Retrospectively, there are three types of vaccines that have been developed against *R. rickettsii*: whole-killed bacteria, subunit vaccines and live-attenuated vaccines. Although whole-killed rickettsiae have not shown apparent toxicity, these inactivated vaccines failed to provide complete protection against RMSF in either animals or humans [23,28,76]. Compared to inactivated whole-organism vaccines, subunit vaccines generally cause less adverse reactions, but they are less immunogenic so that adjuvants are often used with subunit vaccines in order to enhance and modulate the immunogenicity of the antigens. Of course, adjuvants are also useful for whole organism vaccines. Subunit experimental vaccines against rickettsioses have been studied previously and shown to only provide limited or moderate protection against rickettsioses in vivo [77,78,79]. Some surface cell antigens (Sca) in rickettsial species have been described to be involved in participating in adhesion to and invasion of mammalian target cells [80,81,82]. These antigens, including OmpA (Sca0), OmpB (Sca5), Sca2, and Sca4, evolved under positive selection and are present in the genomes of most rickettsial species.

Live-attenuated vaccines usually have the advantages of single dose, rapid onset of immunity, and durable protection, although these vaccines often cause safety concerns.

## 6. Prospects for an Effective Subunit Vaccine

Potential development of a subunit vaccine requires critical evaluation of candidate antigens for evidence of efficacy and stimulation of protective immunity. Surface-exposed proteins such as autotransporters as vaccines for spotted fever group rickettsiae, particularly outer-membrane proteins A (OmpA) and OmpB, which are also known as Sca0 and Sca5, respectively, have been investigated. A non-purified N-terminal fragment of recombinant OmpA that contains conformational epitopes that are reactive with a monoclonal antibody protected mice from the lethal *R. rickettsii* toxicity phenomenon, death within 24 h of intravenous inoculation of an ordinarily lethal dose of rickettsiae [83]. Although this phenomenon of an undetermined, but likely host-mediated, mechanism does not represent an infection, it seems to correlate, although imperfectly, with protection from actual infection [84]. The recombinant OmpA fragment was later shown to also protect guinea pigs from a lethal dose of *R. rickettsii* [85]. Subsequently immunization with baculovirus full-length recombinant *R. rickettsii* OmpA administered with incomplete Freund’s adjuvant followed by one booster immunization protected guinea pigs against an ordinarily lethal *R. rickettsii* challenge [86]. Studies of other species of *Rickettsia* offer further information that is relevant to *R. rickettsii* vaccine design. Immunization with an N-terminal fragment of OmpA of a closely related spotted fever group organism, *R. conorii,* protected guinea pigs from an *R. conorii* challenge and partially protected them against *R. rickettsii* [87]. A DNA vaccine containing a combination of gene fragments of *R. rickettsii sca0* and *sca5* and the corresponding OmpA and OmpB peptides partially protected mice against *R. conorii* infection [87]. Although all of the guinea pigs challenged with *R. rickettsii* survived, eight of nine animals developed fever and five of them lost weight and developed scrotal lesions indicative of disseminated infection. It should be noted that the above studies describe protection, but illnesses often occurred in the animals challenged with virulent organisms.

A critical experiment was immunization of mice with the passenger domain of OmpB (amino acids 36 to 1344) of *R. conorii* produced and purified under native conditions with colonization factor adjuvant and then boosted twice with the same OmpB fragment administered with incomplete Freund’s adjuvant [79]. Most of the immunized mice became ill after retroorbital intravenous challenge and 30% died by day 10. All the control mice including those immunized with denatured OmpB passenger domain died. Although the *R. rickettsii* retroorbital inoculation mouse model lacks validation, these data indicate that native OmpB stimulates some protection and that nonconformational OmpB does not. Previous studies had demonstrated that the passive transfer of a monoclonal antibody reactive with OmpA or OmpB was protective against lethal *R. conorii* challenge in SCID mice and protected against other spotted fever group rickettsiae in guinea pigs, strongly indicating that humoral immunity to OmpA and OmpB is protective [88,89]. Subsequently vaccination of mice with the passenger domain of *R. rickettsii* OmpB produced and purified under native conditions was shown to protect animals from a lethal *R. r**ickettsii* challenge but not an *R. conorii* challenge [90]. Altogether these experiments demonstrate the importance of species-specific conformational antigens of OmpA and the passenger domain of OmpB in stimulating protective immunity against spotted fever group rickettsiae. Furthermore, it has been demonstrated that OmpB of *R. conorii* peptide contains antigens that stimulate CD8 T cells and that both CD8 and CD4 T cells contribute to protective immunity against *R. conorii* [91,92].

Interestingly, a recent study showed different results in protection conferred by subunit vaccine using dogs as the experimental model of rickettsial infection compared to mouse infection [22]. Furthermore, immunization with subunit vaccine, composed of two immunodominant recombinant antigens including Adr2 and OmpB-4, has been reported to reduce rickettsial infection in murine host challenged with *R. rickettsii* [93,94]. However, recombinant Adr2 and OmpB-4 did not show potency as vaccine candidates in dogs because this subunit vaccine failed to show sufficient protection against challenge with *R. rickettsii* after immunization.

## 7. Prospects for a Live Attenuated Vaccine

In association with World War II, the first live-attenuated vaccine against rickettsial disease, *R. prowazekii* strain Madrid E, was generated by a serial laboratory passage of one of the most virulent rickettsiae, *R. prowazekii* [95]. Only a single nucleotide insertion in the methyltransferase gene in *R. prowazekii* turned a highly virulent rickettsial species to avirulent or low virulence rickettsiae leading to the attempts to utilize *R. prowazekii* Madrid E strain as a live-attenuated vaccine against epidemic typhus. However, further investigations demonstrated the failure of *R. prowazekii* Madrid E strain as a vaccine due to reversion of the point mutation leading to restored virulence [96,97,98].

*Rickettsia amblyommatis* is maintained in a large portion of *Amblyomma americanum ticks*, one of the most prevalent and aggressive human-biting ticks in the US. *Rickettsia amblyommatis* is proposed to have much less virulence than *R. parkeri*. However, the actual pathogenicity of *R. amblyommatis* has never been investigated in humans. Interestingly, the geographic distribution of *A. americanum* ticks infected with *R. amblyommatis* is associated with a high prevalence of antibodies to SFGR in healthy people [40]. *Rickettsia amblyommatis* infection induces an asymptomatic immune response that protects against lethal challenge by *R. rickettsii* in a guinea pig model. Therefore, the potential of *R. amblyommatis* as a live attenuated vaccine could be evaluated including its virulence and phenotypic features of the infection in humans. *Rickettsia parkeri* is a low virulence rickettsial species of the spotted fever group, phylogenetically closely related to *R. rickettsii*. Recently, a *Rickettsia parkeri* mutant RPATATE_0245:pLoxHimar (named 3A2) was generated by inserting a modified pLoxHimar transposon into the gene encoding a phage integrase protein [99]. The safety, immunogenicity and efficacy of this rickettsial mutant was evaluated as a live-attenuated vaccine candidate. The safety profile of *R. parkeri* 3A2 was assessed by murine host responses to immunization followed by challenge. Although it is not clear how a phage integrase is important for the virulence of a rickettsial species, interruption of the gene (RPATATE_0245) encoding the phage integrase markedly attenuated the virulence of wild type *R. parkeri* phenotypically. More importantly, this proof-of-concept study demonstrated the feasibility that immunization with one vaccine candidate can confer complete protection against two experimental murine lethal rickettsial infections.

One of the obstacles to developing a vaccine against canine infection with *R. rickettsii* is our gap in understanding of vaccine-induced memory immunity against subsequent rickettsial infection. Fortunately, in two validated animal models, the key elements of host immune protection against primary infection have been demonstrated. TLR4, MyD88, ASC inflammasome, dendritic cells and NK cells have been shown to contribute significantly to a potent innate immunity and to induce an educated and primed adaptive immune response characterized by cytotoxic- and IFN-gamma-producing CD8+ and CD4+ T cells [100,101,102,103,104,105,106,107]. Effector cytokines including IFN-γ, TNF-α, and IL-12 are expected to play a crucial role in host clearance of rickettsial infection in vivo [108,109,110,111]. In line with findings regarding vaccine-induced immunity against pathogens other than rickettsiae, a robust IgG antibody response specific against *R. rickettsii* is indispensable for protection against RMSF. Although we speculate that an immune response, induced by a successful vaccine and required for conferring complete protection against severe RMSF, would be consistent with protective immunity against primary infection, a deeper understanding of vaccine-induced immunity against *R. rickettsii* would facilitate the development of a licensed vaccine. The complete protection against lethal infection with *R. parkeri* or *R. conorii* conferred by *R. parkeri* mutant 3A2 results from both potent *Rickettsia*-specific antibody response and Th1 effector cytokines in sera [99].

## 8. Scope of Rickettsial Mutants for Vaccine Development

Genetic manipulation of rickettsial genomes for generating gene knockouts has been a challenge owing to their obligately intracellular lifestyle, lack of appropriate plasmids and protocols of transformation, restrictions on use of antibiotics for selection, and laborious processes involved in clonal purification of the mutant strains. Despite these bottlenecks, progress has been made in the field, and a few rickettsial mutants have been generated using random (transposon mutagenesis) and site directed (homologous recombination, Targetron and FRAEM methods) mutagenesis approaches (Table 1), and a few of these mutants were tested in animal models of infection to determine their role in virulence.

As obligately intracellular pathogens, *Rickettsia* species enter the host cells via induced phagocytosis, escape from phagosomal vacuoles, and grow in the nutrient-rich intracytoplasmic environment. Several rickettsial proteins including patatins (*pat1* and *pat2*), hemolysin C (*tlyC*), and phospholipase D (*pld*) were implicated for their roles in endosomal escape and establishment of infection in the cytosol [112]. To further ascertain the role of *pld*, a gene deletion mutant was generated by homologous recombination in *R. prowazekii* strain Madrid Evir, and both *R. prowazekii pld* mutant and wild type parental strain were evaluated for virulence in a guinea pig infection model. The animals infected with *pld* mutant (10^6^ pfu) remained afebrile and gained weight while those infected with wild type *Rickettsia prowazekii* developed fever which peaked at day 9–10 p.i., indicating that *pld* mutant is attenuated when compared to the wild type organism. Further, challenge infection of *pld* mutant (10^9^ pfu)-immunized guinea pigs with virulent *R. prowazekii* strain Breinl conferred effective immune protection, and animals remained healthy and gained weight [113]. Recently, *pat1* was shown to have patatin-like phospholipase A2 (PLA_2_) enzyme activity required for evasion of autophagy and escape from the vacuole after host cell invasion. A transposon mutant of *R. parkeri pat1* was avirulent when tested in IFN-I and IFN-ɣ double knockout mice. Although challenged animals lost weight, the majority of mice infected with *pat1::Tn* survived for 40 days post-infection while mice infected with wild type succumbed to infection and died by day 8 p.i. [114]. There was no report of evaluation of the mutant stimulating immunity to virulent challenge.

Surface exposed outer-membrane proteins and adhesins play a pivotal role in rickettsial adhesion and entry into host cells. Among the well-studied and -characterized outer-membrane proteins, while OmpB is present in both spotted fever (SFG) and typhus group (TG) rickettsiae, OmpA is restricted to only SFG species. Although initial studies implicated a role for OmpA in adhesion of *R. rickettsii* to host cells [115], targeted deletion of the gene did not impact the internalization of the mutant strain in vitro, and no significant differences in virulence were observed between the mutant and wild type strain in vivo [116]. Interestingly, *R. parkeri* OmpB mutant (*ompB^STOP^*::tn) despite being serum resistant and able to survive in the blood for 6 h ex vivo was avirulent and could not be detected in organs at 2 to 72 h.p.i in an animal model of infection, indicating a role for OmpB in establishment of infection in vivo [117]. A similar observation of reduced infectivity of host cells by *R. typhi* was observed when OmpB protein expression was disrupted using peptide nucleic acids [118].

Transposon mutagenesis of *R. conorii* and screening of mutants identified disruption of two genes (RC0457 and RC0459) belonging to the polysaccharide synthesis operon. Both the mutant strains exhibited defects in O-antigen synthesis and failed to induce antibody detected by Weil–Felix serology. The mutants were further investigated for their role in virulence in vivo, and animals infected with either wild type or RC0459 mutants exhibited similar levels of weight loss and lethal outcome. However, mice infected with RC0457 mutant did not succumb to infection and remained healthy. Additionally, both mutants did not elicit IgG antibody responses against LPS as observed during the infection with the wild-type strain [119].

The intracellular spread of *Rickettsia* species belonging to the SFG is facilitated by the activation and recruitment of host actin machinery, and rickettsial RickA and surface cell antigen 2 (Sca2) proteins have been functionally characterized for their role in actin polymerization and recruitment of Arp2/3 complex for formation of actin tails and motility. Interestingly, a transposon mutant of *sca2* in *R. rickettsii* not only failed to recruit actin and generate actin comet tails in vitro, but also was involved in virulence in vivo. The *sca2::Tn* mutant despite replicating sufficiently to cause seroconversion did not cause fever in infected guinea pigs as was observed in animals infected with the wild type *R. rickettsii* strain R [120]. Additionally, *sca2* and *rickA,* though required in actin polymerization, were not involved in dissemination in ticks [121]. Recently, a negative regulator of actin tail formation (RoaM) was identified in *R. rickettsii* by transposon mutagenesis. Deletion of RoaM drastically increased actin tail formation but did not influence the expression of genes *rickA*, *sca2* and *sca4* that are involved actin recruitment and polymerization. Additionally, the *roaM* mutant and wild type strains elicited similar host responses and disease in guinea pigs, indicating a minimal role, if any, for the deleted gene in virulence [122].

A few effector proteins secreted by types I and IV (T4SS) secretions systems have been identified in *Rickettsia* species and tested for their roles in virulence. The rickettsial ankyrin repeat protein 2 (RARP-2) secreted by T4SS and involved in fragmentation of host cell trans-golgi network encodes 10 ankyrin repeats in the virulent *R. rickettsii* strain Sheila Smith (SS) while the avirulent Iowa strain harbors a 588-bp deletion within the coding gene and contains only three ankyrin repeats. Expression of full-length SS-RARP-2 in avirulent Iowa strain resulted in restoration of the lytic plaque phenotype typically present with virulent strains; however, Iowa strain carrying SS-RARP-2 was avirulent in guinea pigs [123,124]. Though studies with *R. parkeri sca4* transposon mutant deciphered the role for the gene in binding to vinculin and promotion of protrusion engulfment, no in vivo studies were performed to determine its role in virulence [125].

Himar1 transposon-based mutagenesis of *R. parkeri* strain Tate’s Hell resulted in the deletion of a gene (RPATATE_0245) that encodes a phage integrase family protein. The mutant strain (3A2) despite exhibiting similar levels of growth compared to wild type organisms, resulted in a small plaque phenotype suggesting a role for the deleted gene in virulence. The mutant strain was avirulent; mice injected with the mutant did not develop hepatic necrosis, as observed in mice infected with the wild-type strain. Further, the mutant strain was barely detectable in the organs of infected mice at 4 days post-infection despite inducing strong immunogenic type-I cellular and B-cell responses comparable to those observed in mice infected with the wild-type strain. Most interestingly, mice immunized with a single dose of the mutant demonstrated complete protection against challenge inoculations with wild type *R. parkeri* and *R. conorii*, thus providing proof of concept for utilizing live attenuated rickettsial strains/mutants for the development of vaccines against rickettsioses [99].

However, further studies aimed at functional and immunological characterization of rickettsial mutants in large animal models such as dogs are necessary to evaluate their role in virulence and protective immunity.

## 9. Summary

Breaking the zoonotic cycle that maintains *R. rickettsii*-infected *Rh. sanguineus* ticks in homes and results in transmission of Rocky Mountain spotted fever to the human inhabitants could be accomplished by development of an effective canine vaccine. Two excellent candidate approaches are vaccines that contain live attenuated *R. rickettsii* or a multiplex subunit vaccine containing OmpA and OmpB of *R. rickettsii.* There are vaccine platforms for expression of conformational proteins such as recombinant adenovirus and messenger RNA.

## Figures and Tables

**Table 1 vaccines-10-01626-t001:** List of rickettsial genes disrupted (knock-out or knock down) and tested for their role in virulence and immune protection.

*Rickettsia* Species	Gene	Outcome	Immunization & Protection	Reference
*R. prowazekii*	*pld*	Virulence of the mutant was attenuated in vivo	Yes	[113]
*R. rickettsii*	*sca2*	Mutant strain did not elicit fever in a guinea pig model of infection	No	[120]
*R. montanensis*	*rickA* ^a^	Knock down of RickA was shown to reduce the infectivity of host cells, in vitro	No	[118]
*R. typhi*	*sca5* ^a^	Knock down of Sca5 was shown to reduce the infectivity of host cells, in vitro	No	[118]
*R. rickettsii*	*sca0*	No change in virulence	No	[116]
*R. parkeri*	*sca4* ^a^	Sca4 was shown to bind to vinculin and promote protrusion engulfment, in vitro	No	[125]
*R. parkeri*	*rickA and sca2* ^b^	Both the genes were involved in actin polymerization but not involved in dissemination in ticks.	No	[121]
*R. rickettsii*	*Rickettsia* ankyrin repeat protein 2 (*rarp-2*) ^c^	The expression of RARP2 from Sheila Smith in Iowa did not restore virulence.	No	[123]
*R. conorii*	*RC0459*	No change in virulence	No	[119]
*R. conorii*	*RC0457*	Virulence of the mutant was attenuated in vivo	No	[119]
*R. parkeri*	*sca5*	Virulence of the mutant was attenuated in vivo	No	[117]
*R. parkeri*	Phage integrase family protein	Virulence of the mutant was attenuated in vivo	Yes	[99]
*R. parkeri*	Patatin-like phospholipase (*pat1*)	The *pat1* mutant exhibited reduced virulence in Ifnar1^−/−^ and Ifngr1^−/−^ double knock out mice.	No	[114]
*R. rickettsii*	*A1G_06520* (RoaM [regulator of actin-based motility])	Disruption of the gene increased the number of actin tails. There was no increase in virulence by the mutant in guinea pig model.	No	[122]

^a^ Only in vitro studies were performed. ^b^ The mutants were studied for their functional role in *Amblyomma maculatum* ticks. ^c^ The full length RARP2 protein from virulent strain (Sheila Smith) was expressed in avirulent Iowa strain for in vivo studies.

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
