# Peer review of "A Vaccine for Canine Rocky Mountain Spotted Fever: An Unmet One Health Need"

_vaccines, 2022, doi:10.3390/vaccines10101626_

Round 1
Reviewer 1 Report
In this manuscript titled “A vaccine for canine Rocky Mountain spotted fever: an unmet One Health need”, the authors briefly introduced the background information of Rocky Mountain spotted fever, then reviewed the current development of vaccines against Rickettsia rickettsii. Overall, this is an interesting and well written manuscript. However, this review lack some important details. Therefore, a major revision is needed.
Major issues:
1. The authors started this review with the “One Health” concept without any introduction. The authors need to briefly introduce what “One Health” means for readers not familiar with this concept. Besides, the authors seem to solely focus on vaccines in this manuscript, which is just one part of “One Health”.
2. Line 217-219, “subunit vaccines generally cause less adverse reactions, but they are less immunogenic so that adjuvants are often used with subunit vaccines in order to enhance and modulate the immunogenicity of the antigens”, this description is not accurate. Adjuvants exist in all kinds of vaccines, even in whole killed bacteria. Please specify which adjuvants were used in current subunit vaccines for R. rickettsii.
3. Line 330-333, references are needed.
4. Currently, there is no summary for the whole review at the end of this manuscript. Besides, the authors seem to focus on only two types of vaccines in this manuscript, subunit vaccine and live attenuated vaccine. It would be nice if authors could provide some insights about whether other formats of vaccines could be developed for R. rickettsia in the future in the summary section.
Reviewer 2 Report
This is a very timely Review of the Vaccine issues for a prevalent and important group of understudied human and animal pathogens. The manuscript is very detailed, comprehensive and up to date. Despite the public health importance of RMSF and other associated SFG rickettsial pathogens as the title of this manuscript implies the efforts to develop safe and efficacious vaccine is still unmet. The manuscript as written is OK, however, some segmental reorganization makes it read better. Two other issues came to mind include the descriptive nature of the review (not much data presented or a model provided), and the missed main objective of the manuscript:” A vaccine for canine Rocky Mountain spotted fever. Why there is a need and what is proposed to meet this objective is completely missed in this manuscript. Deletion of the “canine” fix my concern.
Few Specific items:
1. Line 84: delete “will” after D. variabilis.
2. Line 166: what is “unusual hosts”?
3. Line 247: how partial?
4. Lines 302-305: need reference.
Round 2
Reviewer 1 Report
Regarding issue No. 4, the authors still only discussed two vaccines, live attenuated and subunit vaccines. Do authors believe that other types of vaccines will not work for this infection?
All other concerns resolved.
Author Response
We do not believe that the other type of vaccine which has been tested previously, whole killed organism vaccine, will work as it has failed in the past. An mRNA vaccine would be another version of a subunit vaccine, in my opinion, and would have a good chance of being successful. However, it is not my intention to address all potential platforms.